# Mitochondrial Inheritance in Phytopathogenic Fungi—Everything Is Known, or Is It?

**DOI:** 10.3390/ijms21113883

**Published:** 2020-05-29

**Authors:** Hector Mendoza, Michael H. Perlin, Jan Schirawski

**Affiliations:** 1Department of Biology, Program on Disease Evolution, University of Louisville, Louisville, KY 40292, USA; hector.mendoza@louisville.edu; 2Matthias-Schleiden Institute/Genetics, Faculty of Biological Sciences, Friedrich-Schiller University, 07743 Jena, Germany

**Keywords:** mitochondria, replication, inheritance, *Saccharomyces*, *Cryptococcus*, *Ustilago*, *Microbotryum*, human, uniparental, biparental

## Abstract

Mitochondria are important organelles in eukaryotes that provide energy for cellular processes. Their function is highly conserved and depends on the expression of nuclear encoded genes and genes encoded in the organellar genome. Mitochondrial DNA replication is independent of the replication control of nuclear DNA and as such, mitochondria may behave as selfish elements, so they need to be controlled, maintained and reliably inherited to progeny. Phytopathogenic fungi meet with special environmental challenges within the plant host that might depend on and influence mitochondrial functions and services. We find that this topic is basically unexplored in the literature, so this review largely depends on work published in other systems. In trying to answer elemental questions on mitochondrial functioning, we aim to introduce the aspect of mitochondrial functions and services to the study of plant-microbe-interactions and stimulate phytopathologists to consider research on this important organelle in their future projects.

## 1. Introduction

Mitochondria are essential organelles in most eukaryotic cells [1]. They ensure the supply of energy in the form of ATP that can be used for cellular processes. Energy for metabolic reactions is gained from substrates that phytopathogenic fungi take from live or freshly killed plant material. Substrate acquisition will depend on environmental conditions that are highly variable and constantly change with the growth of the fungus. Biotrophic fungi that spread within live plants will face different conditions depending on the invaded tissue, which poses high demands on proper mitochondrial functioning. Mitochondrial function is highly dependent on nuclear gene expression, since most of the mitochondrial proteins are encoded in the host cell nucleus [2]. Despite this high dependence, mitochondria are also highly independent and behave like selfish replicative elements [1]. They possess their own genomic DNA (gDNA) that replicates independently of the nuclear cell cycle, is subject to higher mutation rates, and encodes specific tRNAs for the translation of mitochondrially-encoded proteins. Mitochondria are also highly dynamic. They may occur as egg-shaped compartments in the cytoplasm or they may fuse to form a filamentous network spanning the whole cell [1]. The interplay and regulation of transition between these different forms must be coordinated and integrated so that cellular physiology can respond to rapid changes in the needs of a cell. 

Regulation of these processes requires the action of evolutionarily conserved activities governing mitochondrial fusion and fission, as well as motility and tethering of mitochondria to the cytoskeleton. During cell division, whole mitochondrial organelles are inherited by the new cell. In addition to the mechanical process of distribution between mother and daughter cells, mitochondrial inheritance is regulated in many fungi, and often associated with mating type, being inherited from one parent only. In cases with biparental inheritance patterns, the individual offspring contain mitochondria of the one or the other parent, and heteroplasmy (the occurrence of different types of mitochondria in the same cell) is usually avoided [3]. In this review, we wish to briefly summarize what is known about function and inheritance of these important organelles and how mitochondria pose an additional complex layer of interactions that needs to be considered to fully understand fungal plant infection. In order to foster research in this area, one aim of this review is to highlight knowledge gaps in the field of how mitochondria affect plant pathogen interactions. 

## 2. Mitochondria Are Critical for Cellular Energetics

Mitochondria are essential organelles of eukaryotic cells that provide the cell with chemical energy in the form of ATP. ATP is generated by the cooperation of several protein complexes that form higher supramolecular structures called respirasomes and function in the inner mitochondrial membrane to oxidize NADH and to transfer electrons to a final electron acceptor, most often oxygen [4]. These respirasomes are highly conserved among eukaryotes and are comparable to the membrane-bound electron transport systems found in bacteria and archaea that may use nitrate, fumarate or oxygen as the final electron acceptor [5]. Electron transport results in a transfer of protons across the inner mitochondrial membrane into the intermembrane space, which generates an electrochemical gradient (proton motive force). The accumulated protons flow back to the mitochondrial matrix along the concentration gradient and through a membrane-spanning enzyme complex called ATP synthase that uses the flow of protons for the generation of ATP [6].

Respirasomes of some eukaryotes contain alternative components that compensate for lack or non-functionality of classical complexes of the eukaryotic electron transport chain. One such alternative component is NADH dehydrogenase that is capable of oxidizing extracellular NADH. NADH dehydrogenases were first identified in higher plants [7] and later in fungi [8,9]. A combination of internal and external alternative NADH dehydrogenases is present in *Saccharomyces cerevisiae*, where complex I is completely absent [10]. Alternative components of the mitochondrial respiratory chain can be found in many pathogenic fungi and their further identification and characterization is necessary for the development of novel therapies [11].

Proper mitochondrial functioning is extremely important, and a defect in mitochondrial function can result in serious disease. In humans, a multitude of mitochondrial disorders has been described that cover genetic disorders occurring at birth due to mutations of nuclear genes, as well as those that are based on mutation of mitochondrial genes and that can occur at any age [12]. Some nuclear mitochondrial genes have a role in common diseases like Parkinson’s disease, where a defect in the PTEN-induced kinase 1 (PINK1) or in the E3 ubiquitin ligase, Parkin, lead to a lack of mitochondrial quality control [13]. Symptoms induced by a lack of mitochondrial function are heterogeneous and vary from patient to patient, however, some clinical features are common among affected humans. These include ptosis (dropping eyelids), external ophthalmoplegia (weakening of eye muscles), proximal myopathy (muscle weakness of upper or lower limbs), cardiomyopathy (weakness of heart muscles), and exercise intolerance among others [12]. 

Mitochondrially encoded genetic diseases usually do not occur at birth, and affected individuals carry a mixture of healthy and mutated mitochondrial genomes [14]. The presence of different mitochondrial genomes within the same cytoplasm may result in the dissemination of deleterious mutations arising from the individual nature of each mitochondrial genome (e.g., different DNA replication rates, increased susceptibility to oxidative damage, etc. [15,16,17,18]. The concentration of the mutation needs to exceed a certain threshold level to manifest a disease [19], and this level may vary from person to person and from tissue to tissue [20] explaining the varied clinical phenotypes of individuals affected by a mutation in the mitochondrial DNA (mtDNA) [12]. Mitochondrial heteroplasmies and copy number variations have also been associated with several cancers, including hepatocarcinomas, lung cancers, and breast cancers [16,17,21,22]. Determination of the exact cause of a nuclearly-encoded genetic mitochondrial disorder is also challenging, because more than 1000 mitochondrially located proteins are encoded in the nuclear genome. The nuclear genome controls essential processes needed for proper functioning of mitochondria, like mtDNA maintenance, mitochondrial protein synthesis, coenzyme Q10 biosynthesis and assembly of respiratory chain complexes [12]. 

Mitochondrial malfunction also has a pivotal role in regular senescence processes occurring during aging [23]. When cells age and stop replication but are still metabolically active (i.e., they are senescent), their mitochondria may become more and more dysfunctional, leading to an increased production of reactive oxygen species [24]. At the same time, an increase in reactive oxygen species leads to damage of mtDNA, mitochondrial membrane potential and an overall loss of mitochondrial function [25]. In fungi with a limited life span, senescence has also been attributed to a loss of mtDNA integrity [26]. In plants such as *Arabidopsis*, mitochondria have been shown to stay intact until the very last stages of senescence. During developmental leaf senescence, mitochondria seem to play an important role in orchestrating central metabolic and energetic processes [27].

If mitochondrial defects can be tolerated to some extent in multicellular organisms, how about in unicellular eukaryotes? In unicellular obligate aerobic eukaryotes, mitochondrial defects are rarely tolerated because the resulting fitness defect of the cell will lead to its rapid extinction. Under controlled laboratory conditions however, such mitochondrial mutants can be cultivated and studied. A famous example involves the *S. cerevisiae* petite mutants that form much smaller colonies on nutrient agar than their wild-type ancestors [28]. The petite mutants lack all (*rho^0^*) or a large portion (*rho^-^*) of their mtDNA. Such mutants lose their ability to respire, as well as their ability to grow on ethanol or glycerol as sole carbon sources.

While other cell organelles might be equally important for a proper functioning of the eukaryotic cell, no other cell organelle has so profoundly shaped the early evolution of eukaryotes. So, where do mitochondria come from?

## 3. Mitochondria Have a Long History of Cellular Co-Evolution

Today’s mitochondria are the result of several billion years of co-evolution within eukaryotic cells. Originally proposed in 1967 by Lynn Sagan Margulis, the emergence of the mitochondrion was a direct result of the need to adapt to an oxygen-containing atmosphere [29]. With the establishment and advancement of phylogenetics and mitogenomics, it was recognized that mitochondria are related to a now extinct alpha-proteobacterial lineage [30,31], a conclusion that allowed further development of the endosymbiont hypothesis generally accepted today.

Despite being a ubiquitous organelle in eukaryotes, the mitochondrion still retains a certain level of sovereignty, as it contains its own independently replicating genetic material, protected by a double membrane of comparable composition to the cellular membrane. mtDNA contains protein-encoding genes for key components that make up the respirasome. This apparent independence from nuclear DNA (nDNA) is further accentuated by numerous reports of alternative mitochondrial genetic codes across species [32]. Nevertheless, compatibility between nuclear-encoded and mitochondrially-encoded components of this organelle remain critical for proper function.

The transition from symbiont to mitochondrion involved gradual gene loss and acquisition events. For instance, proteomic studies comparing the ancient symbiont with modern alpha-proteobacteria reveal the loss of proteins involved in replication, transcription and cell division, while retaining those involved in protein synthesis and energy conversion [33]. Perhaps the most exciting finding regarding the evolution of mitochondria is phylogenomic evidence revealing the presence of genes encoding flagellar components, suggesting that the pre-mitochondrion symbiont was free-living [34,35]. Such changes would have generated intermediate states of the organelle-to-be differing in metabolic capacity, until reaching the most suitable levels of synergy with its eukaryotic host.

mtDNA genomes across species also present vast diversity in size and structure, ranging from a mere ~6000 bp in *Plasmodium falciparum* [36] to an impressive ~7 Mb, organized as circular-mapping chromosomes in *Silene noctiflora* [37]. The mitochondrial genome of *S. cerevisiae* is much larger than that of mammals (85 kb versus 16 kb), however, the coding potential is almost the same. The extra sequences mainly stem from the existence of extensive introns and intron-encoded endonucleases. Given this remarkable heterogeneity of the genome of an organelle that is found in almost all eukaryotes [38] and that is derived from a single common ancestor, it can be safely presumed that the most important symbiotic relationship of life itself was not harmonious after all.

The metabolic niche of mitochondria renders mtDNA more vulnerable to mutations than nDNA and this vulnerability is exacerbated by the lack of efficient repair machinery and the simple plasmid-like architecture of mtDNA. Mutations can happen simultaneously in different mtDNA molecules and thus, can rise in frequency and outcompete wild-type mtDNA. This is often referred to as the “tragedy of the cytoplasmic commons”, in which rapid and efficient replication of mutant mtDNA benefits the individual organelle genome but may represent a detriment for the general welfare of the cell [39]. As previously mentioned, generation of a heterogeneous genomic population is linked to the emergence of mitochondrial disorders; however, it has been previously reported to occur at low frequencies (microheteroplasmy) in healthy yeast cells [40] and *Drosophila* [41] and may be even linked to aging processes in humans [42].

## 4. Mitochondria Vary Greatly in Number and Form

With the exception of a few cell types (e.g., mature red blood cells of mammalians or phloem cells of plants) and of a few organisms (e.g., a species of the *Monocercomonoides*, where the essential mitochondrial functions have been replaced by a bacterial-like cytoplasmic sulfur mobilization system [38] and a parasite of salmon, *Henneguya salminicola*, that lacks a mitochondrial genome [43]) eukaryotic cells contain mitochondria. But how many mitochondria are in one cell? For certain cell types and temporal states this question can be answered. However, determined numbers greatly vary (from zero to millions), depending on the type of method used for mitochondrial quantification, the type of organism studied, the type of tissue investigated, the nutritional and developmental state of the investigated cells and the stress experienced by the cells under investigation [44]. 

The study of arbuscular mycorrhizal fungi (AMF) has identified external factors that may shape the fungal mitochondrial density and activation in cells. AMF represent one of the oldest symbiotic associations in the fossil record that may have originated from an initially parasitic or saprobic relationship between higher plants and fungi [45,46]. This interaction depends on plant-derived compounds shown to be of utmost importance for the establishment of endosymbiosis. Evidence of this interaction was first detected by microinjecting fungal hyphae with root exudates, a treatment that stimulated hyphal branching in a dose-dependent manner [47], successfully replicating what is observed when the fungus is grown in the presence of a live root system. Induction of fungal branching is concurrent with expression of mitotic factors [48] and, more interestingly, genes related to mitochondrial function [49]; these include activation of detoxifying enzymes (e.g., P450 complex proteins that degrade phytoalexins), presumably to prevent the plant from killing the fungus during the initial interaction, as well as increased mitochondrial activity, as shown by elevated O_2_ consumption and mitochondrial biomass restructuring [49]. 

The so-called “branching factor” first characterized in the parasitic weeds *Striga* and *Orobanche* as seed germination stimulants [50], belongs to a group of plant sesquiterpenes also known as strigolactones (SLs) that are directly linked to the induction of hyphal branching in different species of AMF [51,52]. Moreover, studies using the synthetic SL analog, GR24, showed that initial branching of AMF during establishment of symbiosis with the rhizosphere triggers a mitochondrial metabolic switch, in which a spike in NADH concentration, NADH dehydrogenase activity and ATP levels are evident [53]. These initial changes are concomitant with active oxidative metabolism, but upregulation of genes related to mitochondrial activity and distribution was only observed after the symbiotic relationship was well established, thus emphasizing the importance of mitochondrial activation prior to mitochondrial proliferation. This finding suggests that SLs are essential for the establishment of a “pre-symbiotic” fungal state, in which mitochondrial activity rapidly increases. GR24 has also been used to study the effects of SLs on phytopathogenic fungi, having both inhibitory and bolstering effects on fungal growth in a variety of fungal species [54], although response to reactive oxygen species could be a critical feature in SL effects on phytopathogenic fungi [55]. Since no clear pattern was observed for all phytopathogenic fungi, it is suggested that SLs were originally used as a primary layer of defense against pathogens and that AMF have evolved an accommodation that utilizes the plant-derived signal to initiate symbiosis through stimulation of a mitochondrial response. 

In addition to environmental and/or host influences, the shapes and sizes of mitochondria can greatly vary not only between different species but also within one species during different developmental processes. Therefore, mitochondrial number is regulated by fission and fusion events and is tied to developmental processes like the cell cycle. A dynamic fission and fusion process determines the number of mitochondria per cell at any given time. This dynamic is essential for the maintenance of mitochondrial structural integrity, function, and appropriate distribution within the cell. The process is required in order to optimize, among other things, the energy output for a given environmental context. Mitochondrial morphology is therefore also highly dynamic. Depending on the cell type or state of the organism, mitochondria may be small oval-shaped organelles, short tubules or even networks spanning the whole cell. They are continuously dividing and fusing, thereby exchanging both soluble and membrane components (e.g., DNA, protein, lipids). The fusion process allows for communication between mitochondrial compartments, which can shield against transient defects in mitochondrial function [56]. In parallel, fission provides a mechanism for the transport, distribution, and quality control-mediated degradation of the organelle [56]. 

The core machinery for mitochondrial division is evolutionarily conserved and involves dynamin related proteins, whose GTPase activity provides for membrane conformational changes. The dynamin related GTPases Drp1 (mammals) / Dnm1p (yeast), and Dyn2 are involved in fission, whereas fusion is promoted by the GTPases Mfn1, Mfn2 (mammals) / Fzo1 (yeast), and OPA1 (mammals) / Mgm1 (yeast). 

For mitochondrial fission, the Dnm1p/Drp1 protein is essential. Dnm1p is normally found in the cytosol and must be recruited to the mitochondrial outer membrane for division [57]. Dnm1p has been shown to form spirals (of approx. 100 nm) that can bind, constrict, and fragment mitochondrial liposomes *in vitro* [58]. The presence of such helices only at later stages of mitochondrial constriction suggested the roles of additional proteins that assist in the process of pinching off or severing membranes [56]. 

For *S. cerevisiae*, additional components of the complex involved in mitochondrial fission include Fis1p, Mvd1p, and Caf4p [56,59]. Fis1 is a mitochondrial outer membrane protein with a single transmembrane segment. Its main protein part is oriented towards the cytosol and it has been shown to recruit the Mvd1p adaptor protein [60]. Mvd1p ensures binding of Dnm1p and the assembly of Dnm1p spirals that encircle and divide the mitochondrial compartment [61]. In contrast, Caf4p has a relatively minor role in Dnm1p assembly. In *S. cerevisiae*, Dnm1p has been shown to interact with Num1p, a protein originally known to be necessary for nuclear migration. Both Num1p and Dnm1p are required for proper mitochondrial inheritance [62]. An additional outer membrane protein, Mdm36p, helps to stabilize the Dnm1p/Num1p interaction and links mitochondria to the cell cortex [63]. Association with the cortex may play a role in mitochondrial inheritance being equal during cell division, since without Dnm1p and Num1p, mitochondria all go to the daughter cells and none remain in the mother cells [62,63].

In mammals, Fis1p is not required for recruitment of Drp1 [64], and human Fis1p (hFis1p) mediated-mitochondrial fragmentation occurs in the absence of Drp1 and Dyn2 [65]. Instead, hFis1 has been shown to bind to and inhibit the GTPase activity of Mfn1, Mfn2, and OPA1, the dynamin related GTPases necessary for mitochondrial fusion. Thus, hFis1 supports mitochondrial fission by blocking the fusion machinery [65].

The fusion of the mitochondrial outer membrane is mediated by the mitofusin proteins Fzo1 (in yeast) and Mfn1 and Mfn2 (in mammals). These proteins also mediate the mitochondrial cross talk with other mitochondria and with other organelles [66]. Fusion of the inner mitochondrial membrane is brought about by Mgm1 (in yeast) and Opa1 (in mammals). Opa1 exists in eight isoforms that are proteolytically cleaved into longer membrane-anchored forms and shorter soluble forms found in the intermembrane space [67]. Through its decisive role in mitochondrial fusion, Mgm1/Opa1 is important for regulating the number of mitochondria per cell.

Mitochondrial fusion and fission events are intimately tied to the cell cycle. During the G1/S transition of the cell cycle mitochondria elongate and fuse to form a hyperfused giant mitochondrial network, whereas mitochondria fragment at the onset of mitosis [68]. Mitochondrial hyperfusion can be induced by inhibition of Drp1 and coincides with a buildup of cyclin E that normally regulates the cell cycle progression through G1/S [68]. In contrast, mitochondrial morphology is not associated with a specific nuclear division state in the filamentous fungus *Ashbya gossypii* [69]. In *A. gossypii*, mitochondria exhibit substantial heterogeneity in both morphology and membrane potential within a single multinucleated cell. Heterokaryons with wild type nuclei and nuclei lacking the mitochondrial fusion/fission genes *DNM1* and *FZO1* exhibit altered mitochondrial morphology. This suggests that in *A. gossypii* the gene products may be required locally near their expression site rather than diffusing widely in the cell [69].

## 5. Mitochondrial Distribution during Cell Division Is Tightly Controlled 

During cell division, mitochondria are partitioned between mother and daughter cells, which requires their association with cytoskeletal elements. In some fungi, such as *Aspergillus* or the budding yeast, or in plant cells, mitochondrial motility is largely actin-based [70]. In yeast, some mitochondria are retained at the base of the mother cell distal to the bud (the “retention zone”), where they colocalize with actin cables [71]. Interaction with actin cables is also necessary for movement of mitochondria into the daughter bud [71]. Indeed, though mitochondria are actively transported along the actin cytoskeleton to the growing bud site in a Myo2-dependent manner, the tethering of specific mitochondria to either the mother or daughter cell is required to ensure proper distribution. An interesting aspect of this process is that daughter cells maintain a constant mitochondria-to-cell size ratio, as well as having “younger” mitochondria, implying communication leading to mitochondrial biogenesis that is coordinated with cell growth [72]. In fact, the mitochondria in the daughter cells have higher redox and membrane potentials than mitochondria retained in the mother cell; the mitochondria in daughter cells also have lower levels of superoxide dismutase. Such age asymmetry is disrupted when Mmr1, the bud-specific tether for Myo2-mitochondria interaction, is missing [73,74].

In mammals, mitochondria are transported to the cleavage furrow during cytokinesis in a microtubule-dependent manner [75]. An association of mitochondria with astral microtubules of the mitotic spindle during cytokinesis was observed via super-resolution microscopy. Dominant-negative mutants of KIF5B, the heavy chain of kinesin-1 motor, and of Miro-1, the evolutionarily conserved mitochondrial Rho GTPase, abrogated mitochondrial transport to the furrow [75]. However, even in systems where microtubules are the primary means of long-distance mitochondrial transport, the actin cytoskeleton is required for short-distance mitochondrial movements and for immobilization of the organelle at the cell cortex [70].

In addition to proper distribution and localization, mitochondrial contact with other organelles is critical for appropriate communication, usually via a tether-mediated contact that influences the behavior or function of one or both organelles [76]. Among the many intracellular contacts made by mitochondria with other organelles, important ones include mitochondrial positioning by dynein anchoring at the plasma membrane during development and differentiation. Mitochondria also make contact with endosomes for iron transfer, and vacuoles and lysosomes for possible lipid and amino acid transport and for mitochondrial division. Finally, contact with the endoplasmic reticulum (ER) is involved in mitochondrial fusion/fission, mtDNA replication and distribution, autophagy, lipid transport, and Ca^2+^ transfer [76]. Indeed, the ER-mitochondria contacts have an effect on the processes influencing both mitochondrial outer and inner membranes during division. 

Integrity of mitochondria and their DNA is provided, not only via fusion-fission dynamics and proper transport, but also by mitophagy and genetic selection of functional genomes [77]. Mitophagy is mediated by autophagosomes, whereby the mitochondria are first segregated in preparation for their degradation in lysosomes. The goal of this process is to remove defective mitochondria, especially under certain cellular conditions. This is distinct from the bulk autophagy response, where functional and dysfunctional mitochondria are indiscriminately removed. This occurs, for example, during nitrogen starvation in *S. cerevisiae* [78]; on the other hand, if cells starved for nitrogen are maintained on a carbon source requiring mitochondrial function, mitophagy is reduced, but not autophagy.

Defects in mitophagy can lead to disease. Mutations in the *PINK1* and *Parkin* genes are associated with familial Parkinson’s disease. Parkin protein, an E3 ubiquitin ligase, is downstream of PINK1 in the pathway in *Drosophila*. In mammalian cells, PINK1 is required for Parkin to translocate from the cytosol to the mitochondrial outer membrane, leading to polyubiquitination of outer membrane proteins. An additional component of this process is played by the AAA ATPase p97, thought to help extract proteins from the outer membrane and make them accessible to the proteasome [77].

Another important factor influencing the distribution and transmission of mitochondria is genetic selection. While genetic selection should be influenced by the relative functionality of individual mitochondria, other factors may also contribute to selection. A severe bottleneck of mtDNA distribution appears to be operating during oogenesis such that only a small percentage of mtDNA molecules winds up in the mature egg. Examples from mice, cows, and humans [77] show rapid segregation of heteroplasmic mtDNA haplotype ratios. Even in cases where a mother has low heteroplasmy, offspring can be produced that are homoplasmic for the rare genotype [79]. One possible explanation for such bottlenecks has to do with the independence of mtDNA replication from the overall cell cycle. Thus, a few mitochondrial genomes in the developing oocyte could be preferentially replicated, whether or not such selective replication is biased towards particular haplotypes.

## 6. Mitochondrial Genome Stability Is Affected by Processes during and after mtDNA Replication 

mtDNA is replicated within the mitochondria by a special set of nuclear-encoded proteins. mtDNA replication is best understood in mammals. Mammalian mtDNA is a small (16.6 kb) double-stranded circular DNA molecule. One of the two strands is richer in guanine bases, making it possible to differentiate a heavy from a light strand. mtDNA replication is mainly carried out by the nuclear-encoded mitochondrially-located DNA polymerase γ (POLγ). While at least four additional polymerases have been shown to also have a role in mitochondria, none of them can replace POLγ [80]. POLγ works together with the helicase TWINKLE that unwinds the double strand in front of POLγ and the mitochondrial single-stranded DNA-binding protein mtSSB that protects the single-stranded DNA against nucleases [81]. The human mtDNA carries two replication origins, O_H_, for initiation of heavy strand replication, and O_L_, located 11 kb downstream of O_H_, where light strand replication is initiated. Replication starts at O_H_ and proceeds by strand replacement, generating a long stretch of single-stranded template that is only replicated once the replication fork has unwound the sequences at O_L_. Initiation of light strand replication starts by the action of the mitochondrial RNA polymerase POLRMT that starts to synthesize a very small transcript that is then used by POLγ for light strand synthesis [81]. RNA primers will later be removed by RNASEH1. When POLγ has completed one round of replication and meets with the DNA strand that it has just polymerized, it starts a process called idling, where it goes into rounds of alternating 5′-3′ exonuclease and 3′-5′ polymerase activities. Idling is necessary for strand ligation by DNA ligase III. Without the 5′-3′ exonuclease activity the replicated strand will be replaced, and circularization of the DNA molecule cannot take place [82]. Daughter molecules may be intertwined during or after replication, and such hemicatenanes need to be resolved by the mitochondrial topoisomerase Top3α [83].

mtDNA replication in yeast follows a different mechanism to that in mammals. For one, enzymes important for replication in mammals are absent from the yeast genome. The yeast genome does not contain genes for a mitochondrial primase, a ribonuclease H or a topoisomerase. Therefore, it seems that replication of yeast mtDNA does not depend on the existence of classical replication origins but is origin-independent. Recent findings led to a reanalysis of previous data and favored a model by which replication of mtDNA in yeast is primed by recombinational structures and proceeded by rolling circle replication and template switching. This allows replication of both leading and lagging strands independent of specific origins and various replicative components needed during conventional DNA replication [84].

mtDNA is packed into compact nucleoprotein complexes visible as mitochondrial nucleoids that usually contain one DNA molecule each [85]. The main protein component is the high mobility group (HMG)-box domain-containing protein TFAM that binds DNA non-specifically via two binding sites, at a density of about one protein per 16–17 bp of mtDNA [86]. Compact and less compact nucleoids exist in each mitochondrion, showing that DNA replication and transcription of mtDNA is not synchronized within mitochondria. Less compact nucleoids have been observed to occur preferentially at sites of mitochondrial contact to the ER. These mitochondria-ER contact sites have been identified as the sites where mitochondrial division takes place, linking mitochondrial division to mtDNA replication. This may ensure an even distribution of newly replicated mtDNA molecules within the mitochondrial network [81,87].

Replication of the mtDNA via POLγ is very precise, since POLγ is a polymerase with proofreading activity that has a low inherent error rate of less than 10^−6^ [88]. However, in *Rhynchosporium* species the mitochondrial mutation rate was found to be higher than the nuclear mutation rate by more than 70 times [89]. The higher increased mutation rate of mtDNA could be due to an increased concentration of reactive oxygen species that are generated at the mitochondrial membrane, and mutations are likely due to an inefficiency of the mtDNA repair system [90]. Point mutations in mtDNA will accumulate with age and an increase in mtDNA mutations may also play a vital role in the aging process [90]. A second type of common mutations in mitochondrial genomes are rearrangements and deletions that could result from errors in replication, double strand breaks or a shortage of repair mechanisms [90]. Since deletions invariably remove essential regions whose loss cannot be tolerated by the cell, these deletions usually occur in heteroplasmy.

Another phenomenon with a potential impact on mitochondrial genome stability is the accumulation of mobile genetic elements, including plasmids (both nuclear-associated and mitochondrial-associated) as well as double-stranded RNA elements [91]. In the broad-host-range pathogen of agriculturally important hosts, *Rhizoctonia solani*, the occurrence of all these elements has been noted, as well as their apparent abilities to escape cell death responses and systems normally associated with regulation of uniparental inheritance. Recent mitogenomic characterization of *R. solani* AG3 isolates led to the complete sequencing of its mtDNA [91]. The data revealed the largest mitochondrial genome of a phytopathogen to date (235849 bp) with high incidence of intronic regions, homing endonucleases, hypothetical genes and repetitive elements. Together, such elements are suggested to be responsible for both the size expansion and diversity of the *R. solani* mt genomes.

In eukaryotic cells, mechanisms exist that allow mitochondrial quality control. Studies from mice suggest that removal of highly defective mitochondria from oocytes occurs in an age-dependent manner, with mitochondria that bear more deleterious mutations being effectively removed with increasing maternal age. Such purifying selection would require monitoring of both the functional capability and the quality of the respective mitochondrial genomes. Interestingly, the removal of mitochondria whose mtDNAs contained non-deleterious polymorphisms seems to be much less efficient [92]. What is particularly interesting about these experiments is that the mechanism of such purifying selection appears to operate at a sensitivity well beyond the initially observable effects of even some seriously deleterious mutations affecting, for example, oxidative phosphorylation. Some severe mutations do not appear to affect function of this critical process until the effects have accumulated beyond a certain threshold [93]. Perhaps whatever is making the decision on removal of defective mitochondria relies on the increased levels of reactive oxygen species associated with certain mtDNA mutations; such a process might be used in conjunction with targeted mitophagy of individually identified defective mitochondria.

## 7. Mitochondrial Inheritance Is Differently Regulated in Different Organisms

Although many sexually reproducing eukaryotes utilize uniparental mitochondrial inheritance, biparental inheritance of mitochondria has been previously described in other systems such as *S. cerevisiae* but also in diverse animal groups [94]. While exclusive maternal inheritance of mitochondria in humans has been the accepted dogma, paternal leakage of mitochondria and persistent heteroplasmy are not as rare as once thought [95,96]. Some degree of biparental inheritance has been found in a wide array of animals (e.g., mammals, arthropods, fish, birds). Specifically, biparental inheritance has been observed in both intra- and inter-specific matings [94].

A unique feature of sexual reproduction is the prevalence of compatibility between two cellular entities thatwill undergo fusion to produce genetically recombinant progeny that are able to adapt to an unpredictable environment [97]. A crucial aspect of sexual reproduction is the inheritance of organelles, with emphasis on the transmission of the genetic material of mitochondria (and chloroplasts in the case of plants). The process of segregation of genetic material of organelles was first described in 1909 by Correns and Baur, who observed different foliage phenotypes in the progeny of *Pelargonium* crosses. Further crossing of this progeny allowed the conclusion that plastids are segregated from each other during vegetative growth and that they are inherited from only one parental strain [98]. It is now known that most angiosperms employ this system, in which plastid DNA and mtDNA are inherited from the maternal strain [99]. In contrast, gymnosperms have been shown to have maternal inheritance of mtDNA but paternal inheritance of plastid DNA [100].

Correns and Baur’s observations embrace an anisogamous scheme of sexual reproduction, in which there is clear discrimination between gametes (e.g., morphology, size, motility, etc.). Gamete asymmetry represents the first checkpoint by which organelle inheritance can be strictly uniparental. In mammals, large size differences exist between the ovum and the spermatozoon: the diameter of the egg is approximately 10^4^ times larger than the length of a fully mature sperm cell. Accordingly, there is a striking disparity in cytoplasmic contributions. Alternatively, isogamous systems, in which there is no clear distinction between sex cells, may also present pre-determined patterns of organelle inheritance. Isogamous systems are suggested to be the ancestral foundation from which sexual reproduction and sex differentiation emerged [101].

Uniparental inheritance is found throughout eukaryotes, ranging from those with similar-sized mating cells, to those with extreme differences in gamete size. The role of uniparental inheritance in selecting against deleterious mutations, as well as in restricting nuclear/mitochondrial and mitochondrial/mitochondrial conflict cannot be overstated. 

The mating type contributing most mitochondria to the next generation is “maternal”, while the other is “paternal”. Control of mitochondrial inheritance can similarly be maternal or paternal. Maternal control involves destruction of the partner’s mitochondria after fertilization, whereas in paternal control, nuclear genes in one mating type control destruction of its own mitochondria during gamete formation. Such controls may be found in unicellular organisms where the mating cells are of similar size, but for multicellular organisms where there is gamete asymmetry, maternal control would amount to the targeting and elimination of mitochondria from sperm post-fertilization. In contrast, under paternal control, the exclusion or disabling of mitochondria occurs during spermatogenesis (before entering the oocyte) [94]. Male organelles are prevented from entering the oocyte in *Ascidian* tunicates, while, in the fungal plant pathogen *Ustilago maydis* (see more below), the maternal (i.e., *a*2 mating type) mtDNA is protected and there is selective elimination of opposite mating type mtDNA after fusion. However, paternal mtDNA may be eliminated without any involvement of the maternal mating type, e.g., in *Drosophila melanogaster*, fish and mice, in which mtDNA is actively degraded during spermatogenesis [102,103,104]. Control can also involve both parents, as seen in bovine and primate sperm where mitochondria are modified with ubiquitin during spermatogenesis, leading to selective degradation after gamete fusion.

In fungi, mitochondrial inheritance has been investigated in some detail in only a few species. However, these investigations provided a large variation in different inheritance mechanisms. Recent results on mitochondrial inheritance mechanisms in *Saccharomyces cerevisiae*, *Ustilago maydis*, *Cryptococcus neoformans* and *Microbotryum violaceum* each provide unexpected solutions to the problem of mitochondrial inheritance as summarized below (Figure 1).

### 7.1. Saccharomyces cerevisiae: Location-Dependent Mitochondrial Inheritance

*S. cerevisiae* is a single-celled eukaryote widely used as a model organism to study the relationships between genes and proteins and extrapolate them into more complex organismal systems. The life cycle comprises haploid and diploid stages, in which reproduction can be achieved through vegetative propagation (budding) or sexual fusion between compatible mating types. The sexual stage of yeast is controlled by the mating type locus *MAT*. Two nonhomologous alleles or “idiomorphs” of this locus, *MATa* and *MATα*, determine the cell mating type, and fusion will only occur between an *a* and an *α* partner. 

Nuclear magnetic resonance and mass spectrometry studies have revealed that the *MATa* cells produce a pheromone (the *a* factor) that carries characteristic post-translationally attached farnesyl residue and a terminal methyl ester group [105], and that *MATa* cells produce cell surface receptors (encoded by *STE3*) specific for α pheromones. Similarly, the *MATα* cells produce the α factor and proteins that activate the expression of cell surface receptors (encoded by *STE2*) specific for *a* pheromones [106]. The *MAT* locus itself encodes regulatory transcription factors, rather than the pheromone or receptor directly. Compatible partners are arrested in the G1 phase of the cell cycle, and pheromone signaling triggers the formation of polarized structural projections, known as “shmoo”, that point towards each other [107]. Time-lapse digital imaging microscopy has revealed that this process is dependent on the coordinate assembly and disassembly of microtubular complexes [108]. 

Successful mating between haploid yeast cells produces a heteroplasmic diploid zygote, which contains a mixture of mtDNA from both parental strains. The *MATa/MATα* diploids are unresponsive to pheromone signaling and will only undergo meiosis to produce four haploid daughter cells through sporulation. Transition to the sporulation program is mediated by repressing *rme1* (repressor of meiosis 1), a haploid-specific gene that normally represses meiosis and promotes mitosis [109]. Distribution of mtDNA among daughter cells is limited to the budding extension point [110,111]. In other words, cells that originate from buds from either end of the mother cell will inherit mtDNA from one parental strain; whereas cells that originate from a midpoint will inherit a mixture (Figure 1). *In vitro* studies have shown that yeast displays mostly biparental inheritance of mtDNA [112]. However, this pattern is lost after approximately twenty rounds of mitotic cell division, restoring and selecting for homoplasmic daughter cells.

An interesting mitochondrial inheritance pattern is observed in the petite mutants (*rho^-^*) of *S. cerevisiae*, that lack large fractions of their mtDNA leading to a respiration deficiency. When wild-type cells are crossed with hyper-suppressive *rho^-^* petite mutants, the progeny are exclusively of the *rho*^-^ mitotype. This observation suggests an extremely biased inheritance pattern for the hyper-suppressive *rho^-^* mitotype [113].

### 7.2. Ustilago maydis: Degradation-Mediated Uniparental Mitochondrial Inheritance

*Ustilago maydis* is one of the best characterized fungal plant pathogens. This smut fungus only infects corn (*Zea mays*) and its progenitor (teosinte) and has allowed the study of recombination, plant-pathogen interactions, mating type loci and, among others, inheritance of mtDNA. Haploid cells (sporidia) of this unicellular eukaryote are generated through meiosis of germinated diploid teliospores and proliferate by budding. For plant infection and pathogenic development, mating-compatible sporidia need to fuse. Fusion and the maintenance of the subsequently formed dikaryotic filaments is controlled by two unlinked mating type loci: the *a* locus that encodes a precursor for a small lipopeptide pheromone and a seven transmembrane G-protein coupled pheromone receptor [114], and the *b* locus that encodes the two subunits of a heterodimeric regulatory protein responsible for infectious development [115].

The *a* locus exists in two non-homologous idiomorphs (*a1* and *a2*), while the *b* locus is multiallelic, existing in at least 23 different forms [114]. Successful mating will only occur between cells of different mating types for both *a* and *b* loci and is initiated by pheromone signaling that leads to the formation of conjugation tubes. Conjugation tubes from mating-compatible sporidia fuse at their tips, which leads to plasmogamy and the formation of stable dikaryotic hyphae. Dikaryotic hyphae proliferate within the plant tissue, lead to the local induction of tumors, and after fusion of the nuclei develop into diploid teliospores that can spread, upon tumor bursting, and germinate under favorable conditions to give birth to haploid sporidia of different mating types.

The study of mitochondrial inheritance was possible by the discrimination of different mitotypes following the identification of a polymorphic region within the LSU rRNA region of mtDNA in different *Ustilago* strains [117]. Crossing experiments of strains with different mitotypes showed that most offspring only contain mitochondria of one of the two parental strains and that the donating parent was of the *a2* mating type, suggesting that in *U. maydis*, uniparental mitochondrial inheritance takes place [117]. The *a2* locus contains two genes not present in *a1, lga2* and *rga2* [118]. To test whether they are involved in the regulation of mitochondrial inheritance deletion strains were created. Deletion of *lga2* resulted in biparental inheritance of mtDNA [117]. Interestingly, the *lga2* deletion also led to the generation of recombinant mtDNA molecules. The inheritance of these novel mitotypes was favored when the strains were of the *a2* mating type. In contrast, deletion of *rga2* favored the inheritance of *a1* mitotypes. Expression of *rga2* in the *a1* mating partner resulted in biparental inheritance of mtDNA [117]. These results are best explained by a model in which the direct or indirect role of Rga2 is protection of mitochondria or mtDNA from the direct or indirect action of Lga2 (Figure 1). Lga2 would have a direct or indirect role in degradation of mtDNA that is not protected by the action of Rga2 [117]. Lga2 was found to interfere with mitochondrial fusion. This process of Lga2-induced mitochondrial fission is in part mediated by the dynamin-related GTPase Dnm1 known to be involved in mitochondrial fission but not by a possible influence of Lga2 on stability of the fusion protein Fzo1 [119].

The discovery of recombination within the mtDNA molecule raises questions regarding other recombination hot spots and at what frequencies they occur. mtDNA recombination has been previously reported in numerous pathogenic fungi and is suggested to be essential for the purging of deleterious mutations from a lineage [116].

The process of uniparental inheritance of mitochondria mediated by active degradation is common among most eukaryotes that promote the inheritance of maternal mtDNA. For instance, in mammals, numerous studies have described that degradation of paternal mtDNA is achieved through ubiquitination of mitochondria in the sperm prior to fertilization [120,121,122]. As a result, upon cytoplasm fusion, paternal mitochondria are targeted for destruction by the proteasome or the lysosome. In *Caenorhabditis elegans*, paternal mtDNA degradation is regulated by autophagy, as revealed by fluorescence microscopy experiments, in which paternal mitochondria are sequestered by autophagosomes upon injection into the egg’s cytoplasm [123]. Mutations in autophagy-related genes led to persistence of paternal mitochondria during late embryogenesis stages. In *D. melanogaster*, degradation of paternal mtDNA was observed to occur during gametogenesis [102]. During tail formation, mitochondria of the sperm cell fuse to form an elongated organelle with multiple nucleoids containing the genetic material. An endonuclease produced during the maturation of the sperm cell targets mtDNA. In this manner, only maternal mtDNA will be inherited in the zygote following fertilization.

### 7.3. Cryptococcus neoformans: Genetic and Physical Constraints during Uniparental Mitochondrial Inheritance

*Cryptococcus neoformans* is a basidiomycete fungus that exhibits a dimorphic lifestyle, being able to switch from a unicellular budding yeast form to filamentous hyphal growth. Sexual dimorphism has been previously linked to the virulent nature of *C. neoformans* based on its ability to produce infectious spores from filamentous hyphae, on the genetic variability that sexual reproduction provides as a basis for increasing fitness and virulence, and on the importance of mating type loci in both mating and infection. *C. neoformans* exists as both a free-living form and in association with a variety of plant and animal hosts. More importantly, this fungus behaves like an opportunistic pathogen when infecting immunocompromised animal hosts. The fungus is the causative agent of cryptococcosis, a defining opportunistic infection of AIDS that may lead to life-threatening meningoencephalitis.

*C. neoformans* mating is regulated by a bipolar mating system, *MAT*, that defines each mating type (a or α) [124]. As it is the case in other heterothallic fungi, mating in *C. neoformans* can only occur between compatible cells. This mating reaction is regulated by pheromone and pheromone receptor genes (P/R) encoded within the *MAT* locus [125]. In addition to pheromone and pheromone receptor genes, the *MAT* locus also contains genes encoding homeodomain (HD) transcription factors Sxi1α and Sxi2a that exert conspicuous regulation over sexual development [126].

Mitochondrial inheritance in *C. neoformans* is predominantly uniparental, as demonstrated by evidence of post-mating hyphal growth with mitochondria originating from the MATa parent only [127]. The uniparental inheritance pattern is also observed in interspecific crosses (*C. neoformans* vs. *C. deneoformans*) [128]. Deletion of either *sxi1α* or *sxi2a* results in mitochondrial leakage from the MATα parent, while exchange of the *sxi2a* gene for the *sxi1α* gene in the MATa parent had no effect, suggesting the involvement of other genes that have yet to be identified [129].

A study in 2013 identified Mat2, a transcription factor not encoded within the *MAT* locus, as being activated in MATa cells during pheromone signaling, and as being involved in tagging mitochondria for preservation by an unknown mechanism [130]. In the MATα parent, Mat2 activation leads to the formation of a conjugation tube towards the MATa parent. Upon plasmogamy, a Sxi1α/Sxi2a complex arbitrates the activation of downstream factors that lead to the elimination of α mitochondria (Figure 1). Mat2 activity can be influenced by environmental factors, leading to a leaky mitochondrial inheritance pattern [131]. Additionally, unisexual crossings between MATα cells have been reported to result in a biparental mitochondrial inheritance pattern [128], suggesting different Mat2 regulation in MATa and MATα cells.

In addition to genetic control mechanisms, organelle inheritance in *C. neoformans* may also depend on physical constraints during mating. For instance, conjugation tube formation is asymmetrical, emerging from the MATα parent and polarizing the soon to be zygote. Subsequent hyphal growth from the zygote always occurs at a site opposite from the conjugation side (the “a” side; Figure 1) [132]. These mechanisms impose physical restrictions on the introgression of α mitochondria into the MATa parent and have been observed in other species like *Agaricus bitorquis* [133] and *Coprinus cinereus* [134].

One possible selective advantage for the uniparental controls of mitochondrial inheritance exhibited by *C. neoformans* may have to do with protection against selfish DNA elements found in mtDNA. As previously mentioned, deletion of either *sxi1α* or *sxi2a* gene led to a biparental inheritance pattern. A recent study revealed that deletion of *sxi1α* also resulted in recombinant mtDNA genotypes associated with *cox1* introns [135]. Since the recombinant *cox1*introns are associated with homing endonuclease genes, mobile genetic elements that may exhibit non-Mendelian transmission frequencies and incur no particular benefit other than their own spread, uniparental control may thereby protect *C. neoformans* offspring from homing endonuclease genes. As such, *C. neoformans* appears to be properly equipped with the molecular armament to prevent the spread of such selfish genetic elements.

### 7.4. Microbotryum violaceum: Doubly Uniparental Inheritance of Organelles

*Microbotryum violaceum* is an obligate pathogen of the carnation family/”Pinks” (Caryophyllaceae) that sterilizes its host by replacing pollen on the anthers of inflorescences with teliospores, hence its moniker “anther smut”. The pathogenic development of this fungus highly resembles that of *U. maydis*, thus its previous categorization as *Ustilago violacea*. Phylogenetic studies, as well as studies based on spore morphology and host specificity, led to the development of the new genus *Microbotryum* [136]. Teliospores of *M. violaceum* are transmitted by pollinators to healthy hosts, followed by germination and meiosis to produce haploid sporidia.

*M. violaceum* has a heterothallic bipolar mating system, in which successful mating of haploid sporidia is determined by a specific mating type locus (*a*). This locus exists in two forms, *a1* and *a2*, and determines the mating type of the parental strains. Upon interaction, the compatible partners will trigger a pheromone reaction and the formation of conjugation tubes. Both *a* forms encode specific pheromone receptors that show significant homology to Ste3 pheromone receptors of *S. cerevisiae* [137]. The pheromone components of the mating locus remain uncharacterized, although synthetic pheromone has been produced and used experimentally [138]. Upon fusion of haploid sporidia, dikaryotic hyphal growth is triggered, allowing the pathogen to penetrate the plant tissue and produce a systemic infection. The fungus overwinters in the meristematic tissue of its host until flowering, at which stage diseased inflorescences emerge. 

Early studies using mtDNA restriction fragment length polymorphisms revealed that *M. violaceum* presents a special type of uniparental mitochondrial inheritance [139]. In these early experiments, progeny of the *a1* type inherited mtDNA from either parental strain, while progeny of the *a2* type only acquired mtDNA from the *a2* parental strain (Figure 1) [139]. The mechanism of this special type of uniparental mitochondrial inheritance, termed doubly uniparental inheritance is not well understood in *M. violaceum* as the necessary molecular and biochemical studies are lacking. However, doubly uniparental inheritance of mitochondria has been previously described in some bivalve mollusks as a process for sexual differentiation [140,141], providing an excellent model to study mitochondrial function in determination and maintenance of sex.

## 8. Conclusions

Mitochondria are ancient organelles with a long history and an elemental impact on evolution. In this light it is not surprising that mitochondrial functions are highly conserved among eukaryotes. A lot of what we understand today about mitochondria, their role, their function, their composition, their morphology, their replication, their organellar interactions, their evolutionary and functional constraints, was elucidated in very few organisms and is assumed to be universal. However, even very basic processes like mtDNA replication differ greatly between mammals and yeast. The greatest variation of mechanisms has been found in mtDNA inheritance and these mechanisms are especially diverse in fungi. For a few proteins their role in the dynamic modulation of mitochondria has been studied, but for the vast majority of mitochondrially located proteins no function is known so far. Even for proteins where a functional role has been elucidated (e.g., for Lga2 of *U. maydis*), the exact function is still unclear. In addition, there are a lot of open questions and unexplored connections that we need to answer in order to fully understand how phytopathogenic fungi colonize, live in, and exist with their plant hosts (Figure 2). We hope that these questions may stimulate the formulation of additional questions that can be studied and answered in different plant-fungal pathosystems so that we ultimately reach a comprehensive picture of the different roles, functions, services, molecular components and their regulation, as well as their impact on plant-fungal interactions of this important organelle.

## Figures and Tables

**Figure 1 ijms-21-03883-f001:**
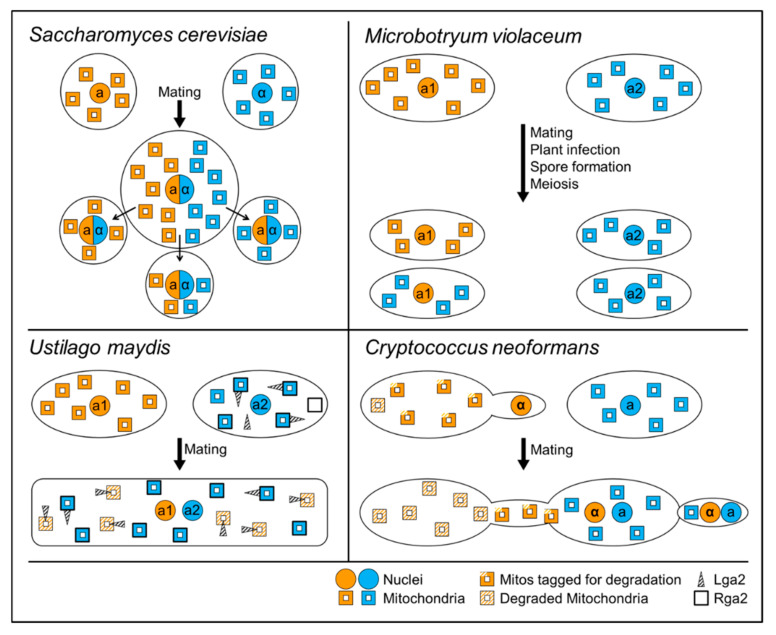
Mechanisms of mitochondrial inheritance in fungi. In *S. cerevisiae*, a and α type mitochondria stay relatively local upon zygote formation. Whether offspring contain a or α type mitochondria or a mixture of both depends on the local position of bud emergence. Figure modified from [91]. In *M. violaceum*, sexual development results in offspring where a1 type cells contain either a1 or a2 type mitochondria and a2 type cells contain only a2 type mitochondria. The mechanism of this doubly uniparental inheritance pattern is unknown. Figure modified from [116]. In *U. maydis*, two proteins encoded in the a2 mating type locus are responsible for uniparental mitochondrial inheritance. Rga2 shields mitochondria from the degradative effect of Lga2. Upon dikaryon formation, Lga2 leads to degradation of unprotected a1 type mitochondria. Figure modified from [96]. In *C. neoformans*, two effects might lead to uniparental mitochondrial inheritance. During pheromone stimulation, α type mitochondria might be tagged for degradation and be degraded during zygote formation. Only α type cells form conjugation hyphae through which the α type nuclei migrate into the zygotes. The zygote buds off at the opposite pole, thus further reducing the chance of inheriting α type mitochondria. Figure modified from [112].

**Figure 2 ijms-21-03883-f002:**
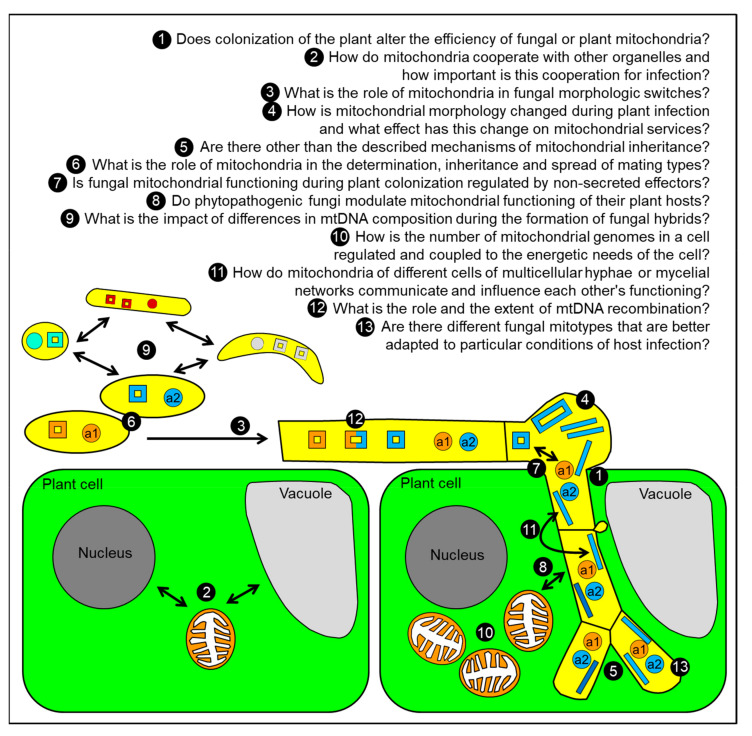
Some open questions on mitochondrial inheritance in phytopathogenic fungi.

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
