# Peer review of "Mitochondrial Inheritance in Phytopathogenic Fungi—Everything Is Known, or Is It?"

_ijms, 2020, doi:10.3390/ijms21113883_

Round 1
Reviewer 1 Report
as requested, I reviewed the manuscript (ID ijms-817962) "Mitochondrial inheritance in phytopathogenic fungi 2 – everything is known, or is it?", by H Mendoza, M H Perlin, and J Schirawski.
This review manuscript deals with the description of mitochondrial inheritance in phytopathogenic fungi. The general topics are focused on mitochondrial roles, their origin, their distribution and genomic features. Then, a section on mitochondrial inheritance finally points at pathogenic species with specific reference.
The overall discussion contains all the elements for comprehension of the subject, for understanding the cited results and for finding hints to deepen information. The reading sounds like a comprehensible continuum, and there’s no evidence of fragmented information. The manuscript is conceived for giving the necessary keywords to enter the investigated subject, and reaches this goal.
On this basis, the authors demonstrated scientific mastery of the topic.
Nevertheless, I add below very minor suggestions for reaching a better final shape.
Minor comments:
- Although a review paper has informative scope, the titles of the sections sound actually scholastic. This is actually a minor comment and is not a mandatory indication, anyway: i) a review paper is not a text book, and ii) style is not negligible in publishing a review paper.
- Sections 4 and 5 might be unified (finding a new concept title). Thus, the total new paragraph might also have 20% reduced length than the 4+5 sum.
- Figure 2: the reviewer’s feeling is that an effort in making a graphical representation (e.g. a scheme) of the mere list could enhance the value of the manuscript, its disclosability and its publishable standard style.
Thank you very much for your attention to my opinion.
Author Response
We thank the reviewer for this constructive criticism.
Following the reviewer’s suggestion, we have changed section headlines from scholastic questions to more informative statements (highlighted in yellow in the manuscript).
We found that combining sections 4 and 5 would not decrease the word count of the article but rather make it less easy to read. We therefore opted for leaving the text separated in two sections but gave new headlines to the sections.
We changed the appearance of Figure 2 and included a graphical representation.
We hope that this adequately addresses the reviewer’s concerns and led to an improvement of the manuscript.
Reviewer 2 Report
Dear authors, the manuscript is very interesting. I ahve little suggestions:
introduction: please, add the citations.
2 and 3. I suggest reversing the order of chapters 2 and 3
you have not discussed anything about senescence; as you know, mitochondria remain intact in plants until the final stage of senescence. And in phytopathogenic fungi?
Moreover, compounds such as strigolactones increase the density of mitochondria of AM fungi . about phytopathogenic fungi?
Finally, why didn't you consider pythium sp. and/or rhizoctonia solani?
best regards
Author Response
We thank the reviewer for these thoughts on how to further improve the manuscript.
We can understand the rationale of changing orders of chapters 2 and 3 (i.e. start with the beginning/the origin), however, we deliberately put the chapter on importance first, in order to stress the many indispensable functions of mitochondria before following the question of origin. Therefore, we decided to leave the chapter order as is.
Following the reviewer’s suggestions, we inserted new references in the introduction.
We now also included a paragraph on senescence, adding to the importance of mitochondria for cellular functioning including references (lines 98-106).
We also included a new paragraph on mycorrhiza and the function of strigolactones on mitochondria (lines 168-198) and included a paragraph on the mitochondrial genome of Rhizoctonia solani (lines 371-380) all with references.
We extended the list of references and changed all reference numbering.
We hope that these changes both improved the manuscript and adequately considered the reviewer’s comments.